# Fractal-Stereometric Correlation of Nanoscale Spatial Patterns of GdMnO₃ Thin Films Deposited by Spin Coating

Yonny Romaguera-Barcelay [1], Ştefan Ţălu [2,*], Robert Saraiva Matos [3], Rosane Maria Pessoa Betânio Oliveira [3], Joaquim Agostinho Moreira [4], Javier Perez de Cruz [5] and Henrique Duarte da Fonseca Filho [1,*]

[1] Laboratory of Nanomaterials Synthesis and Nanoscopy, Physics Department, Federal University of Amazonas, Amazonas 69067-005, Brazil; yonnyroma@gmail.com
[2] The Directorate of Research, Development and Innovation Management (DMCDI), Technical University of Cluj-Napoca, 15 Constantin Daicoviciu St., 400020 Cluj-Napoca, Romania
[3] Department of Materials Science and Engineering, Federal University of Sergipe, São Cristóvão, Sergipe 49100-000, Brazil; amazonianmaterialsgroup@gmail.com (R.S.M.); rosaneboliveira@ufs.br (R.M.P.B.O.)
[4] IFIMUP and Departamento de Física e Astronomia da Universidade do Porto, Rua do Campo Alegre s/n, 4169-007 Porto, Portugal; jamoreir@fc.up.pt
[5] Instituto de Soldadura e Qualidade, Av. Prof. Dr. Cavaco Silva, 2740-120 Porto Salvo, Portugal; jcruz@fc.up.pt
* Correspondence: stefan_ta@yahoo.com (S.Ţ.); hdffilho@ufam.edu.br (H.D.d.F.F.)

**Abstract:** Multiferroic systems are of great interest for technological applications. To improve the fabrication of thin films, stereometric and fractal analysis of surface morphology have been extensively performed to understand the influence of physical parameters on the quality of spatial patterns. In this work, GaMnO₃ was synthesized and thin films were deposited on Pt(111)/TiO₂/SiO₂/Si substrates using a spin coating apparatus to study the correlation between their stereometric and fractal parameters. All films were studied by X-ray diffraction (XRD), where the structure and microstructure of the film sintered at 850 °C was investigated by Rietveld refinement. Topographic maps of the films were obtained using an atomic force microscope (AFM) in tapping mode. The results show that the film sintered at 850 °C exhibited a clear formation of a GdMnO₃ orthorhombic structure with crystallite size of ~14 nm and a microstrain higher than other values reported in the literature. Its surface morphology presented a rougher topography, which was confirmed by the height parameters. Topographic differences due to different asymmetries and shapes of the height distributions between the films were observed. Specific stereometric parameters also showed differences in the morphology and microtexture of the films. Qualitative rendering obtained by commercial image processing software revealed substantial differences between the microtextures of the films. Fractal and advanced fractal parameters showed that the film sintered at 850 °C had greater spatial complexity, which was due to their higher topographic roughness, lower surface percolation and greater topographic uniformity, being dominated by low dominant special frequencies. Our combination of stereometric and fractal measurements can be useful to improve the fabrication process by optimizing spatial patterns as a function of the sintering temperature of the film.

**Keywords:** GdMnO₃; morphology; topography; stereometric and fractal analysis

## 1. Introduction

One of the special classes of materials that has attracted, in recent years, the attention of researchers and industry in general are multiferroics, because they show simultaneous coexistence of ferromagnetic and ferroelectric properties in the same phase. Thus, multiferroic materials have been of great interest in several areas, like spintronics, memory storage devices, sensors, electronic components, among others [1–7]. Moreover, the perovskite oxides of RMnO₃ type (R is rare earth element) are a novel class of materials exhibiting a set of remarkable interesting properties from the physics point of view, such as superconductivity,

ferroelectrics, ferromagnetism, multiferroics and colossal magnetoresistance [8,9]. The simplicity of the perovskite structure turns it into the most common crystalline structure among the studied multiferroic materials. This is due to its fairly simple organization and chemical constitution, with the general formula $ABX_3$, where A and B are cations and X is an anion, usually oxygen, but others, like fluorine, can be used.

Among the magnetoelectric materials, $GdMnO_3$ has been studied as this material exhibits magnetic-induced ferroelectricity [10], although other study proposals are being carried out. As an example, Rasras et al. [11] carried out a study of the magnetic properties, and magnetic memory effect in nanocrystalline single-phase powders of $GdMnO_3$ synthesized by the sol–gel method. Ye et al. [12] prepared $GdMnO_3$ ceramics by solid-state reaction method and analyzed the structure, defects, dielectric, and magnetic properties of those samples sintered at different temperatures. At room temperature, $GdMnO_3$ shows an orthorhombic distorted perovskite structure with Pbnm symmetry [13,14], and it has attracted considerable interest in recent years due to its applications in multifunctional devices, and underlying physics, where its antiferromagnetic and ferroelectric properties are strongly connected. As can be noted, many recent studies focusing on the structural, ferroelectric, and magnetic properties of $GdMnO_3$ have been reported in the literature [11,15–18]. However, it is intriguing and challenging to note that there are rare reports involving the morphological study of its surfaces through the atomic force microscopy (AFM) technique, which is an important tool to evaluate their physical properties that are crucial for advanced technological applications. As mentioned before, the AFM technique is a peculiar technique, due to it being both sensible and accurate and recently, morphological studies have been done through topographical maps, providing several parameters, such as stereometric [19–23], multifractal [24–27], and power spectrum density (PSD) [21,28,29]. All these parameters allow us to characterize surfaces at micro- or nanoscale, showing how the spatial patterns of the film surface, such as roughness, shape of peaks, peak density, heterogeneity of topographic texture distribution among others, are modified as the sintering temperature is varied.

In this work, three $GdMnO_3$ films were synthesized from a specific precursor solution, and deposited by spin coating onto $Pt(111)/TiO_2/SiO_2/Si$ substrates, and thereafter sintered at different temperatures. Morphological analysis, and nanoscale pattern results, along with the combined stereometric and advanced fractal parameters are discussed in detail, which has not yet been reported for the as-processed $GdMnO_3$ films. Much of the presented results were obtained using MountainsMap commercial software. It is worth stressing that the results extracted from fractal data, were carried out by self-developed analytical methods, which are not provided by commercial software.

## 2. Materials and Methods

### 2.1. Material and Sample Preparation

In this work, we have previously dissolved lutetium (III) nitrate hydrate 99.99% pure (supplied by Aldrich) at 50 °C, in a glacial acetic acid ($CH_2CO_2H$) and nitric acid ($HNO_3$) 2:1 molar ratio mixture, for 24 h, to produce a $GdMnO_3$ precursor solution. It was immediately added to a stoichiometric molar content of manganese (II) acetate tetrahydrate (($CH_3COO)_2$ $Mn\cdot x4H_2O$), 99.99% pure (supplied by Merck), and, with pure 2-methoxyethanol, in a solvent (2:1:6) molar ratio ($CH_2CO_2H/HNO_3/CH_2OCH_2CH_2OH$), achieving a 0.2 molar concentration [30]. To produce the thin films used in our study, $Pt/TiO_2/SiO_2/Si$ substrates were spin coated at 3000 rpm for 60 s with the precursor solution using a Laurell WS-400-6NPP instrument. They were dried at 80 °C on a hot plate for 1 min and pre-sintered at 400 °C in a tubular furnace for 10 min, and this process was repeated 9 times [31], being sintered at 650, 750, and 850 °C for 1 h. We labelled these samples as GdMnO650, GdMnO750, and GdMnO850.

### 2.2. Structure Analysis

The structural evaluation of $GdMnO_3$ thin films was made using a PANalytical MRD diffractometer that uses a Cu K$\alpha$ radiation ($\lambda$ = 1.5418 Å) source with a size step of 0.025°/10 s from 10 up to 80° (2θ). Subsequently, the films were analyzed using X'pert Highscore software to identify the phases. Moreover, a structural Rietveld refinement was carried out using the Fullprof package [32] on the sample sintered at 850 °C to obtain the refined structural parameters.

### 2.3. SEM and AFM Measurements

To make a broader analysis of the surface of $GdMnO_3$ thin films, high resolution scanning electron microscopy (SEM) experiments were performed on an FEI Quanta 400 FEG ESEM 3 equipment, using 15 kV at 25 °C.

AFM analysis of the $GdMnO_3$ thin film surface was done using a Veeco Multimode equipped with a NanoScope IVa controller. Areas of $2.5 \times 2.5$ $\mu m^2$ with a scan rate of 1.0 Hz and a resolution of $256 \times 256$ pixels were carried out in tapping mode. In all samples, four measurements were made in random regions along the thin films surface.

### 2.4. Surface Analysis

The stereometric parameters that were the basis for the thin film's morphology surface analysis, were in accordance with the ISO 25178-2:2012 standard, whose parameters have their physical meaning largely described in [33–36]. To compute several parameters, such as height, feature, spatial, functional, hybrid, volume, and core Sk, MountainsMap 8.0 commercial software was used (Digital Surf, Mountains 8.0, (2020). https://www.digitalsurf.com accessed on 5 December 2020). Furthermore, contour lines, furrows, and texture directions, obtained by Fourier transform on the height function, were obtained from surface microtexture evaluation of the films.

The fractal characterization of rough surfaces is one of the many fields of science and engineering that makes use of Mandelbrot fractal mathematics, for example, Weierstrass–Mandelbrot (WM) function [37]. This is because a single mathematical expression (the WM function) contains characteristics that mimic the emergence of roughness. As an example, based on a method using image analysis, Guariglia [38] investigated the link between fractal geometry and prime numbers. Moreover, as wavelets are powerful mathematical tools to analyze 1D/2D signal data in the time–frequency domain, many methods have been proposed based on wavelet and fractal theories, such as image classification using wavelet transform [39]. Thus, in addition to the stereometric parameters, and to have a complete evaluation of the sample surface, we also calculate advanced fractal parameters to study the surface microtexture. According to the method described by Mandelbrot and Wheeler, fractal dimension (FD) was computed using a counting box [40] as well as the fractal lacunarity (FL) which was computed using a model described by Salcedo et al. [41]. From the lacunarity curve, we estimated the lacunarity coefficient (β) using Equation (1), to obtain data regarding the surface texture homogeneity [42].

$$L(r) = \alpha . r^\beta \tag{1}$$

where $L(r)$ is lacunarity, $\alpha$ is a constant, and $r$ is the box size.

Using linearized graphs obtained from the mathematical theory by Jacobs et al. [43], the average power spectrum density (PSD) of fractal regions of the spectra were calculated. Besides, we have estimated the Hurst coefficients of all spectra using Equation (2), from a linearized graph, where $\gamma$ is the slope of the linearized curve, obtained using WSxM 5.0 software [44].

$$Hc = \frac{\gamma - 2}{2} \tag{2}$$

Moreover, using the model described by Melo and Conci [45], fractal succolarity (FS) was calculated using Equation (3).

$$FS(T(k), dir) = \frac{\sum_{k=1}^{n} P_0(T(k)).PR(T(k), p_c)}{\sum_{k=1}^{n} PR(T(k), p_c)} \tag{3}$$

where *dir* is the liquid entry direction, $P_0(T(k))$ is the occupation percentage, $T(k)$ are boxes of equal size $T(n)$, *PR* is the occupation pressure, and $p_c$ is the centroid position $(x, y)$.

Finally, the last parameter named surface entropy (E) was obtained from the information theory using the Shannon entropy equation [46,47], Equation (4), where $p_{ij}$ is assigned to be the probability that a height matrix term $h_{ij}$ promotes a complete uniformity of the height distribution.

$$E^{(2)} = -\sum_{i=1}^{N} \sum_{j=1}^{N} p_{ij}.log(p_{ij}) \tag{4}$$

The obtained value was centralized and normalized according to Equation (5) to give us a normalized value of *E* [48].

$$E = \frac{E^{(2)} - E_{min}^{(2)}}{E_{max}^{(2)} - E_{min}^{(2)}} \tag{5}$$

All advanced fractal parameters were computed from the AFM topographical matrix extracted by WSxM software. The computational routines were programmed in R language (For FS and E) and Fortran 77 (for FL).

### 2.5. Statistical Analysis

For the results presented in this study, we used the analysis of variance (ANOVA) and Tukey test with a *p*-value of 0.05.

## 3. Results

### 3.1. Structural Evaluation

The X-ray diffraction (XRD) patterns used to evaluate the structure of the films are shown in Figure 1. In Figure 1a, the patterns of GdMnO650 and GdMnO750 are shown, where a comparison can be made. Notably, GdMnO650 displays only reflections associated with the substrate, explicitly from the planes (111), (200), and (220) of Pt and (211) of $TiO_2$. The GdMnO750 pattern also has these reflections and an additional one referring to the $SiO_2$ (202) plan. However, GdMnO750 clearly shows the growth of the $GdMnO_3$ perovskite phase, which was confirmed using an Inorganic Crystal Structure Database (ICSD) code collection card #95493. In this regard, it can be observed that the planes (121), (002), (022), (311), and (303) are associated with the peaks positioned at 33.2, 33.9, 41.7, 52.02 and 60.01°, respectively.

The pattern that most consistently displays the phase of interest is shown in Figure 1b, where a Rietveld structural refinement was performed to obtain the main refined structural parameters. A $GdMnO_3$ orthorhombic phase is observed associated with the Pnma spatial group and superimposed with reflections from the substrate is observed. From a resolution file obtained with a LaB6 standard sample, the peaks' widths were obtained using a model of spherical harmonics [32]. The structural refined parameters, lattice parameters, unit cell volume, average crystallite size, strain, and dislocation density are summarized in Table 1. From the obtained lattice parameters a = 5.31715(2), b = 5.78694(3), and c = 7.44901(6), it can be observed that a and c are relatively higher than those reported for a $GdMnO_3$ single crystal [49], a $GdMnO_3$ ceramic [14] and another system similar to ours previously reported [31]. The obtained unit cell volume 229.206(6) Å is greater than the one reported by Romaguerra et al. [31] and less than those reported by Peña et al. [14] and Mori et al. [49]. Furthermore, the crystallite size and microstrain factor of the GdMnO850 lattice have magnitudes of 14.227 (7) and 7.1264 (4), respectively, where the dislocation density, which is a parameter associated with the degree of crystalline solid defects [50], exhibits a value of 0.00494(1) (Table 1). Comparatively, the crystallite size is smaller than the one reported by

Romaguerra et al. [31] and Rasras et al. [11], which is a result of the more nanometric nature of the particles in our films. This behavior affected the configuration of structural defects since it was also observed that the lattice microstrain was considerably larger than the one reported by those authors. Evidently, a greater distribution of defects in crystalline solids is associated with larger microstrains, due to the discordance movement and increased grain boundary along the surface of the material.

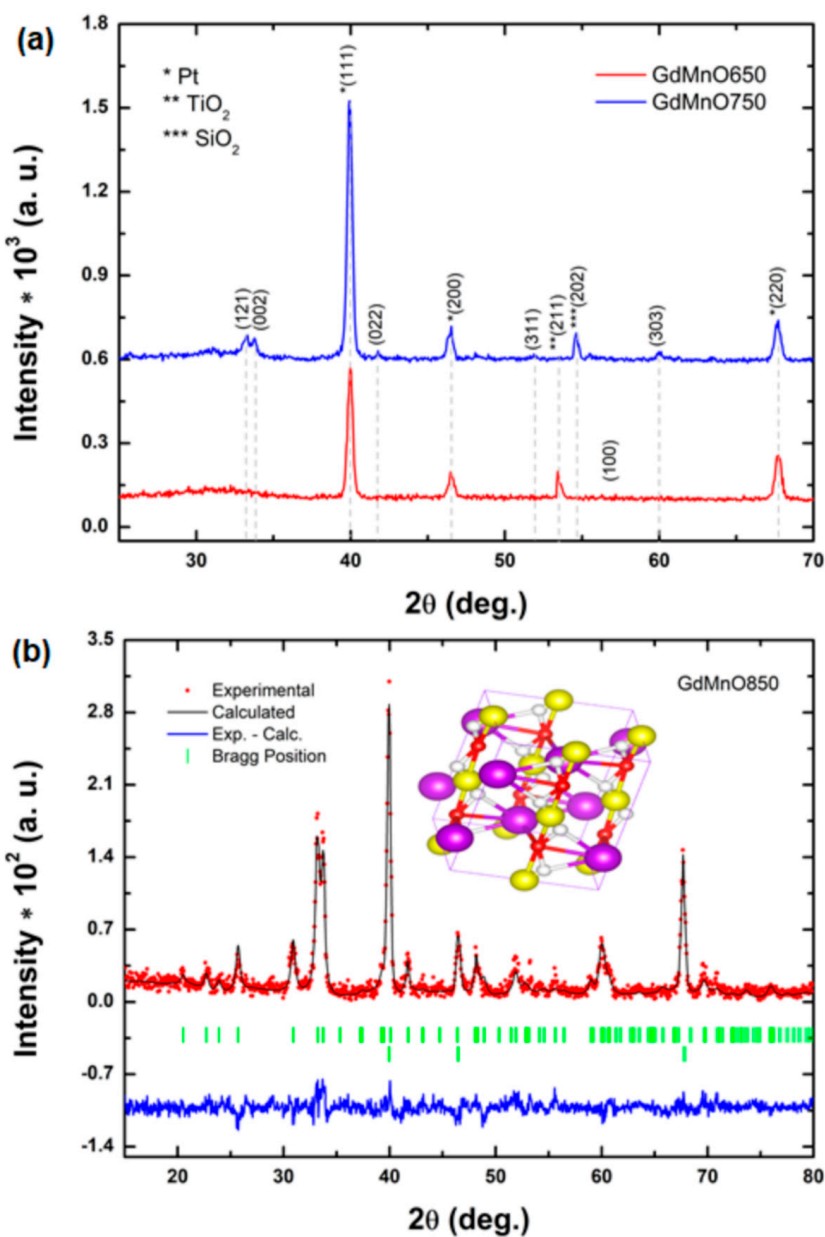

**Figure 1.** (**a**) X-ray diffraction (XRD) patterns of GdMnO$_3$ thin films sintered at 650 and 750 °C. (**b**) Structural Rietveld refinement of GdMnO$_3$ thin film sintered at 850 °C.

**Table 1.** Structural parameters of GdMnO850 thin film obtained by structural Rietveld refinement.

| Sample | a (Å) | b (Å) | c (Å) | $V_c$ (Å$^3$) | $D_{(xrd)}$ (nm) | $\varepsilon$ | $\Delta$ (1/D$^2_{(xrd)}$) |
|---|---|---|---|---|---|---|---|
| **GdMnO850** | 5.31715(2) | 5.78694(3) | 7.44901(6) | 229.206(6) | 14.227(7) | 7.1264(4) | 0.00494(1) |

### 3.2. Surface Morphology Analysis

In order to guarantee the formation of films with adequate thickness, at each deposition cycle, layers of 20 nm were deposited, which at the end of the ninth cycle provided films with a thickness of ~180 nm. After sintering there was no change in thickness as a function of temperature increase. Scanning electron microscopy plan-view images of the $GdMnO_3$ thin films are shown in Figure S1 (Supplementary Material), revealing the evolution of the surface as a function of the increasing sintering temperature. In this figure, we present images with a larger viewing area (Figure S1a,c,e) and others with a greater magnification of the surface of thin films (Figure S1b,d,f). The samples annealed at temperatures of 650 °C (Figure S1a,b) and 750 °C (Figure S1c,d) exhibit smooth surfaces without visible grain outlines, which is characteristic of an amorphous or incipient crystalline phase. However, $GdMnO_3$ films annealed at 850 °C (Figure S1e,f) exhibit grains and a well-defined structure.

Furthermore, 3D nanoscale morphology and a 2D representation of the relative frequency of topographic maps of $GdMnO_3$ thin films are shown in Figure 2. GdMnO650 and GdMnO750 exhibited smoother surfaces (Figure 2a,c). The topography of GdMnO850 (Figure 2e) is more irregular and has a distribution of sharper peaks, which can be attributed to the growth of $GdMnO_3$ grains due to the increase in temperature that promoted the emergence of larger structures. Romaguera-Barcelay et al. [31] reported the evolution of $GdMnO_3$ film morphology through scanning electron microscopy measurements, whose results were similar to the AFM images shown here. Negi et al. [15] reported the fabrication of $GdMnO_3$ multiferroic thin films on $SrTiO_3$ (110) substrate by pulsed laser deposition (PLD) technique and their AFM images showed the granular nature of the film [51]. A complementary 2D observation can be found in Figure S2. These qualitative observations can be confirmed by the increase in the roughness parameters that are shown in Table 1, where all parameters showed significant difference ($p < 0.05$). GdMnO650 and GdMnO750 presented similar roughness values, which were observed for mean roughness (Sa) and standard quantitative statistical parameters of image amplitude (Sq), while GdMnO850 has the highest roughness (~3.6 nm), suggesting greater spatial complexity (greater topographic irregularity). Similarly, the maximum peak height (Sp), maximum pit height (Sv), and maximum height (Sz) also exhibited similar behavior because GdMnO850 presented the highest values (Table 2), revealing that the organized and nanostructured growth of $GdMnO_3$ crystals in GdMnO850 changes the topography of the film, because of the increase of the sintering temperature. Notably, the increase of roughness may be associated with the high microstrain value of the $GdMnO_3$ lattice, when compared to other values reported in previous studies, for example [11,31], as mentioned in Section 3.1.

**Table 2.** Height surface parameters of $GdMnO_3$ thin films, according to ISO 25178-2:2012.

| Parameter | Unit | GdMnO650 | GdMnO750 | GdMnO850 |
|:---:|:---:|:---:|:---:|:---:|
| | | Height | | |
| Sq | (nm) | 1.58 ± 0.25 | 1.65 ± 0.22 | 3.63 ± 0.30 |
| Ssk | (-) | 0.44 ± 0.21 | 0.15 ± 0.27 | −0.08 ± 0.08 |
| Sku | (-) | 3.61 ± 0.42 | 3.48 ± 0.23 | 2.63 ± 0.04 |
| Sp | (nm) | 7.80 ± 2.39 | 6.58 ± 1.42 | 17.12 ± 5.49 |
| Sv | (nm) | 4.49 ± 0.25 | 5.31 ± 0.81 | 12.11 ± 0.51 |
| Sz | (nm) | 12.28 ± 2.34 | 11.88 ± 1.59 | 29.22 ± 5.59 |
| Sa | (nm) | 1.23 ± 0.21 | 1.28 ± 0.15 | 2.97 ± 0.26 |

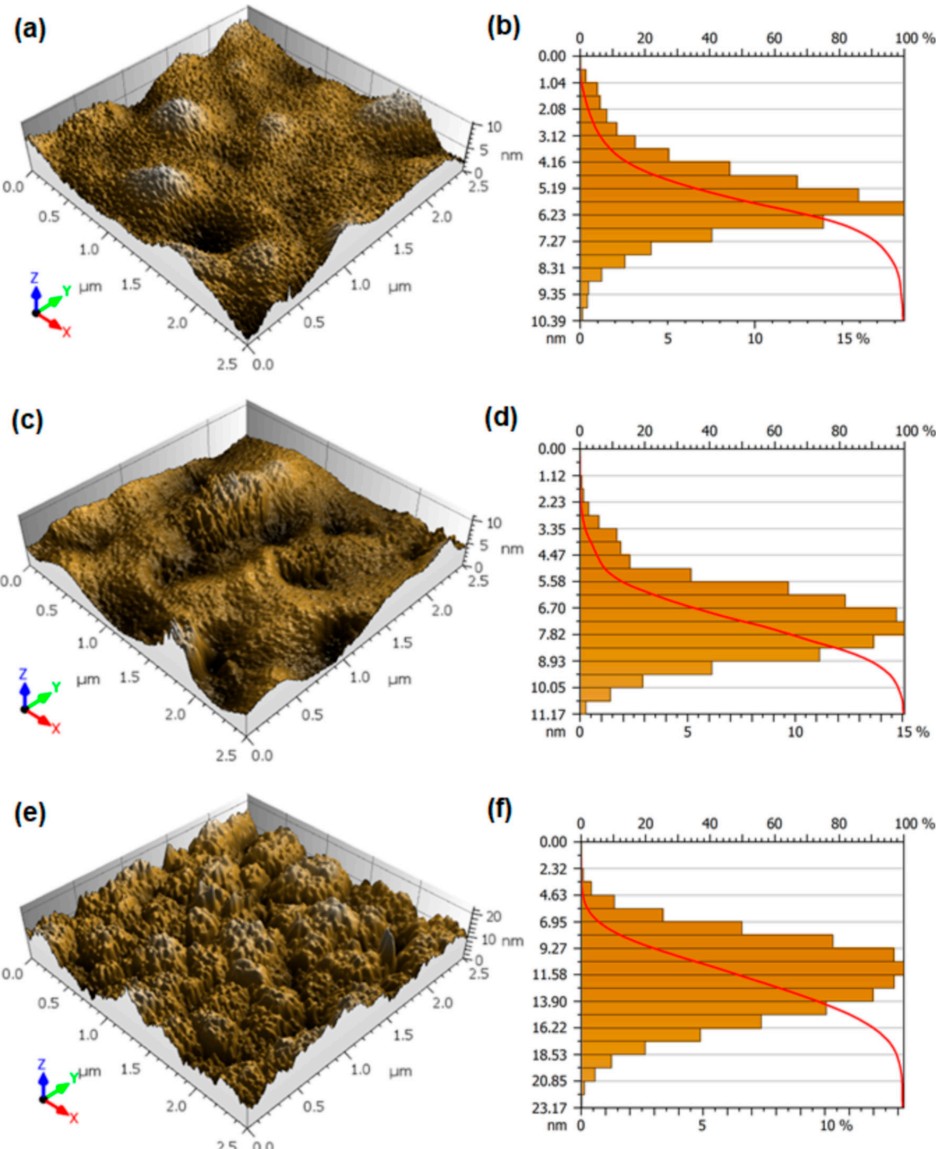

**Figure 2.** 3D atomic force microscopy (AFM) topographical maps and histogram of relative heights of GdMnO₃ thin films of (**a**,**b**) GdMnO650, (**c**,**d**) GdMnO750, and (**e**,**f**) GdMnO850.

Additionally, the combination of the events shown in Figure 2b,d,f, the symmetry parameter Ssk (skewness), and shape of the height distribution Sku (kurtosis) confirm the topographic difference generated by the increase of the sintering temperature. A negative asymmetry was computed for GdMnO850, but as Ssk was ~0, a more symmetrical pattern is assigned to this film [36]. In contrast, GdMnO650 and GdMnO750 showed positive asymmetries and further from the symmetric value (Ssk = 0). Interestingly, all values are still close to 0, suggesting that there is strong topographic uniformity of the films. Moreover, GdMnO650 (~3.6) and GdMnO750 (~3.5) exhibited greater kurtosis than GdMnO850 (~2.6) (Table 2). Such kurtosis values show that the surfaces present different shapes of height distribution, where GdMnO650 and GdMnO750 exhibit more pointed shapes (leptokurtic) and GdMnO850 has a flatter distribution (platykurtic) [36,52,53], confirming the different configurations of the spatial patterns that occurred due to the increased sintering temperature. Moreover, the Abbott–Firestone [54] curves of Figure 2b,d referring to GdMnO650 and GdMnO750 present a smooth S-like shape, which was not observed for GdMnO850 that has no symmetrical curve. This result shows that at frequent depths, the content of the covered material in relation to the evaluated area is greater for

GdMnO650 and GdMnO750 than for GdMnO850, which may be due to the flatter character of GdMnO850.

### 3.3. Advanced Stereometric Evaluation

Several other stereometric parameters associated with the surface of $GdMnO_3$ thin films are shown in Table 3. According to Blateyron [34,35], these parameters, which are obtained from carrying out Fourier analysis to the height autocorrelation function, provide information about the spatial patterns of the material's microtexture. In this regard, almost all functional parameters showed a statistically significant differences ($p < 0.05$). The microtexture of GdMnO850 has a different spatial configuration, where a greater inverse areal material ratio (Smc) was observed for GdMnO850 (~4.7 nm), while GdMnO650 and GdMnO750 exhibited lower values (Table 3). This behavior was also noted for peak extreme height (Sxp), clearly indicating that the topographic structures of GdMnO850 have a different rough profile regarding GdMnO650 and GdMnO750.

**Table 3.** Stereometric parameters of the $GdMnO_3$ thin films, in accordance with ISO 25178-2:2012.

| Parameter | Unit | GdMnO650 | GdMnO750 | GdMnO850 |
|---|---|---|---|---|
| | | Functional | | |
| Smc | (nm) | $2.07 \pm 0.42$ | $2.01 \pm 0.27$ | $4.67 \pm 0.42$ |
| Sxp | (nm) | $2.72 \pm 0.12$ | $3.29 \pm 0.51$ | $7.09 \pm 0.36$ |
| Sk | (nm) | $3.58 \pm 0.69$ | $3.88 \pm 0.31$ | $10.05 \pm 90.01$ |
| Spk * | (nm) | $2.17 \pm 0.51$ | $2.11 \pm 0.59$ | $2.64 \pm 0.30$ |
| Svk | (nm) | $1.33 \pm 0.21$ | $1.75 \pm 0.47$ | $2.93 \pm 0.25$ |
| Smr1 | (%) | $14.02 \pm 1.55$ | $10.47 \pm 1.08$ | $7.64 \pm 0.52$ |
| Smr2 * | (%) | $90.95 \pm 2.01$ | $89.20 \pm 2.07$ | $90.28 \pm 0.38$ |
| Vmp * | $(\mu m^3/\mu m^2)$ | $1.04 \times 10^{-4} \pm 2.15 \times 10^{-5}$ | $1.03 \times 10^{-4} \pm 2.79 \times 10^{-5}$ | $1.41 \times 10^{-4} \pm 1.63 \times 10^{-5}$ |
| Vmc | $(\mu m^3/\mu m^2)$ | $1.31 \times 10^{-3} \pm 2.38 \times 10^{-4}$ | $1.38 \times 10^{-3} \pm 1.53 \times 10^{-4}$ | $3.52 \times 10^{-3} \pm 3.45 \times 10^{-4}$ |
| Vvc | $(\mu m^3/\mu m^2)$ | $2.02 \times 10^{-3} \pm 4.44 \times 10^{-4}$ | $1.92 \times 10^{-3} \pm 2.60 \times 10^{-4}$ | $4.43 \times 10^{-3} \pm 4.24 \times 10^{-4}$ |
| Vvv | $(\mu m^3/\mu m^2)$ | $1.54 \times 10^{-4} \pm 8.26 \times 10^{-6}$ | $1.96 \times 10^{-4} \pm 4.23 \times 10^{-5}$ | $3.80 \times 10^{-4} \pm 1.50 \times 10^{-5}$ |
| | | Feature | | |
| Spd * | $(1/\mu m^2)$ | $31.00 \pm 17.93$ | $10.00 \pm 0.43$ | $20.44 \pm 10.62$ |
| Spc * | $(1/\mu m^2)$ | $11.86 \pm 2.56$ | $8.00 \pm 1.00$ | $10.98 \pm 2.70$ |
| | | Hybrid | | |
| Sdq | (-) | $0.04 \pm 0.00$ | $0.03 \pm 0.01$ | $0.085 \pm 0.002$ |
| Sdr | (%) | $0.07 \pm 0.01$ | $0.05 \pm 0.01$ | $0.35 \pm 0.01$ |

* Samples without significant difference, ANOVA one-way and Tukey test ($p > 0.05$).

As a result of differences in the surface roughness of the films, other differences were also observed regarding the portion of the material that makes up the surface microtexture (thickness) and the volume of the material that fills the representative peaks and valleys of the topographic texture. The graphical representation associated with those parameters can be found in Figure S3, whose full definition can be consulted in [35,36]. The results show that GdMnO850 has the highest (~10 nm) core surface thickness (Sk), while GdMnO650 and GdMnO750 have lower and close values (Table 3). A persistent pattern was displayed for reduced valley depth (Svk), revealing that there are different contours between rough peaks due to the formation of different channels along with the surface microtexture of GdMnO850 in relation to GdMnO650 and GdMnO750. This generated different dale void volume (Vvv), core void volume (Vvc), and core material volume (Vmc) for GdMnO850. Nevertheless, the rough peak shape seems not to have been affected by the sintering

temperature, as the reduced peak height (Spk) and peak material volume (Vmp) showed no statistically significant difference ($p > 0.05$). The 3D maps exhibited in Figure 2 reveal a similar peak shapes for all films.

Feature specific parameters of peak density and peak shape confirm that all films have similar peak shape and spatial configuration. Moreover, peak density (Spd) and arithmetic mean peak curvature (Spc) also showed no statistically significant difference ($p > 0.05$), revealing that although the films have different topographical spatial intensities, the peak structure does not change. However, the hybrid parameters reveal that GdMnO850 has a less flat surface because the root mean square gradient (Sdq) was the highest one. The same behavior was recorded for the developed interfacial area ratio (Sdr), where GdMnO850 presented a greater value in relation to GdMnO650 and GdMnO750. A full flat surface is assigned to Sdq of ~0 [55] and therefore GdMnO850 is less flat, which is naturally due to its greater topographic irregularity (greater spatial complexity).

*3.4. Microtexture Analysis*

MountainsMap software provides qualitative renderings that are also obtained from Fourier transforms to the profile height function. Those renderings simulate furrows and arrange contour lines along the surface that represent qualitative aspects of the surface microtexture, as shown in Figure 3. As can be seen, greater holes (black regions) are observed in GdMnO650 (Figure 3a) and GdMnO750 (Figure 3c) than in GdMnO850 (Figure 3e). More intense valleys (intense color regions) are observed in GdMnO850 that are due to their highest roughness.

There is an overlap of microchannels that form contours between rough peaks. These contours are shown in Figure 3b,d,f and reveal that particle agglomerates are formed in GdMnO650 and GdMnO750., which confirms the lower growth of grains in these conditions. Furthermore, because of the greater roughness of GdMnO850, the rough peaks tapered the channels, which promoted the appearance of deeper channels, as the maximum depth of furrow (~12.2 nm) and the mean depth of furrow (~4.6 cm/cm$^2$) was considerably greater for this sample. This differs from the other samples because there was a statistical difference between the means ($p < 0.05$) (Table 4). However, GdMnO650 exhibited the highest mean density of furrows because there is a greater distribution of furrows in this sample.

**Table 4.** Surface microtexture parameters of GdMnO$_3$ thin films according to ISO 25178-2:2012.

| Parameter | Unit | GdMnO650 | GdMnO750 | GdMnO850 |
|---|---|---|---|---|
| | | Furrows | | |
| Maximum depth | (nm) | 3.38 ± 0.43 | 4.16 ± 0.56 | 12.24 ± 0.52 |
| Mean depth | (nm) | 1.51 ± 0.13 | 1.34 ± 0.19 | 4.60 ± 0.34 |
| Mean density | (cm/cm$^2$) | 86,382.81 ± 4997.06 | 78,881.94 ± 1442.56 | 81,635.01 ± 564.91 |
| | | Texture | | |
| TI | (%) | 63.58 ± 10.78 | 40.45 ± 15.08 | 65.44 ± 7.54 |
| Str | (-) | 0.64 ± 0.11 | 0.40 ± 0.15 | 0.65 ± 0.08 |
| Std | (°) | 169.75 ± 4.68 | 141.19 ± 7.05 | 120.62 ± 19.88 |

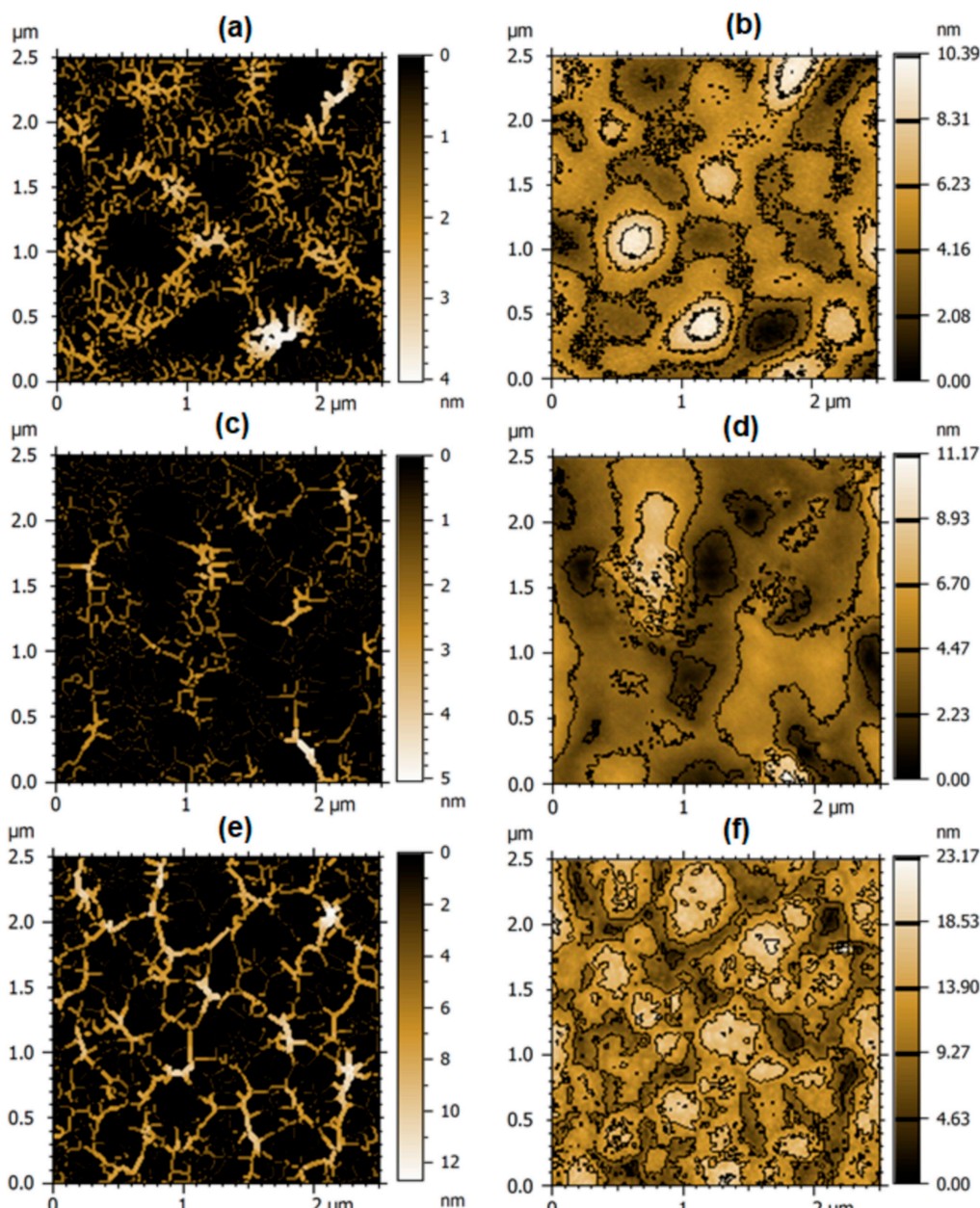

**Figure 3.** Qualitative rendering of furrows and contour lines systems of the surface microtexture of (**a,b**) GdMnO650, (**c,d**) GdMnO750, and (**e,f**) GdMnO850.

Notably, the different topographic profiles generated by the different sintering conditions of the thin films influenced the film's microtexture, as well as the texture isotropy, as displayed in Figure 4. As can be seen, there was a consistent smoothing of the texture distribution of GdMnO650 to GdMnO850, which is due to platykurtic behavior (Sku < 3), but a greater dispersion within the distribution of GdMnO850 was observed, which is attributed to the rougher profile of this surface. Although GdMnO650 and GdMnO850 have similar isotropies and texture aspect ratios (Str) ~64 and 65% and 0.64 and 0.65, respectively, the texture distributions are different, where GdMnO850 presents the lower mean angle (~121°) for general texture direction (Std) (Table 4), which occurred due to the gradual change in surface microtexture from GdMnO650 to GdMnO850.

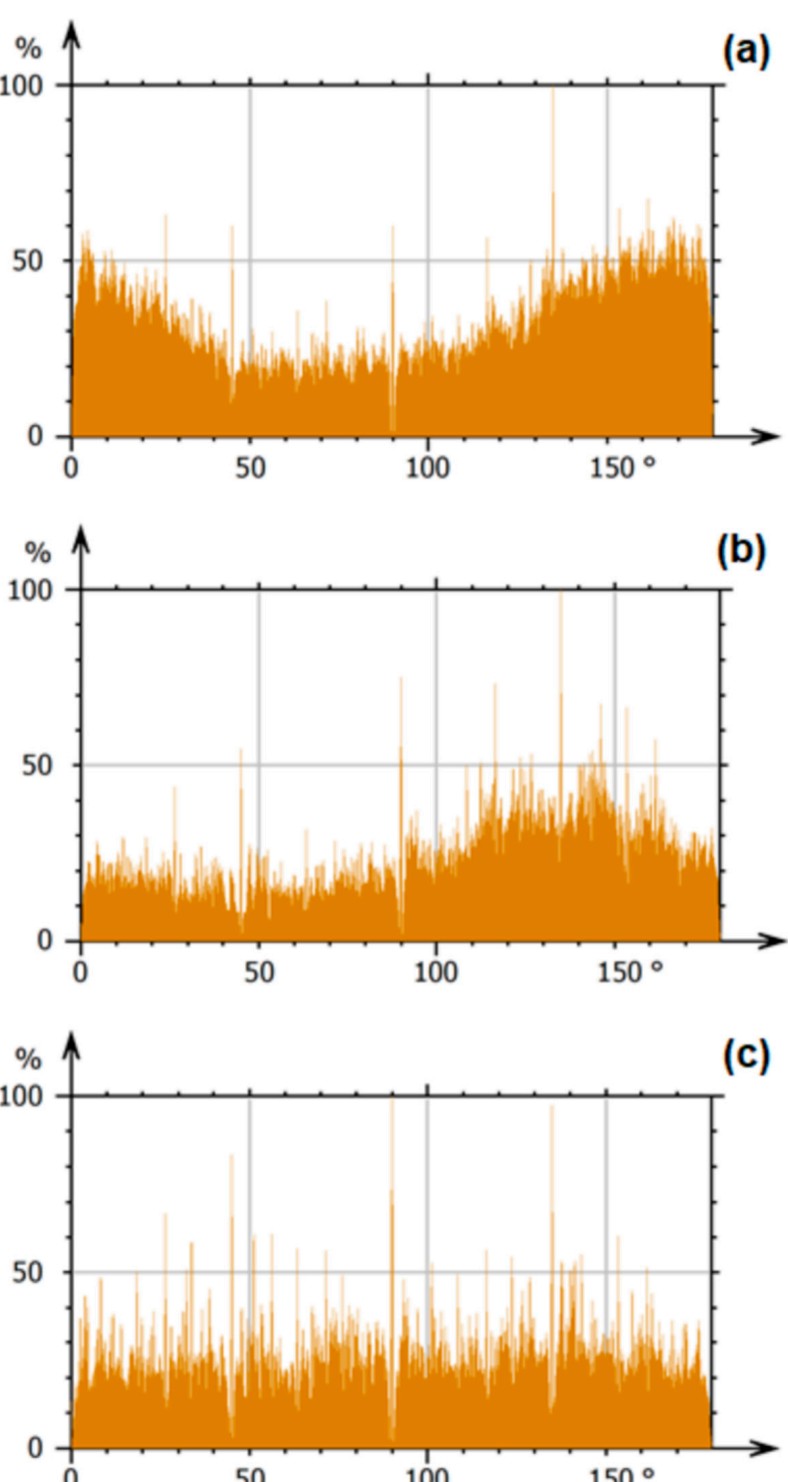

**Figure 4.** Texture directions of surface microtexture of (**a**) GdMnO650, (**b**) GdMnO750, and (**c**) GdMnO850.

### 3.5. Fractal Characterization

Fractal and advanced fractal tools are very useful modern approaches used for the characterization of thin films and others systems, for example [23,56–59]. Hurst coefficient (Hc) and fractal dimension (FD) are well-known parameters used to extract information about the spatial complexity of thin films, for example [19,21,29]. However, new advanced fractal parameters such as fractal lacunarity (FL), fractal succolarity (FS), and surface entropy (E) were recently used by Talu et al. [42], which expanded the analysis about surface spatial complexity. In this way, Figure 5 shows the power spectrum density

and lacunarity distribution of the fractal region of representative experimental curves of each analyzed film. As can be observed in Figure 2a–c, the surfaces presented self-affine behavior, where well-adjusted fits were recorded. The Hurst coefficients are displayed in Table 5, where all fractal parameters exhibited a statistically significant difference ($p < 0.05$). There was a larger difference between samples, as GdMnO650 and GdMnO750 exhibited lower values of Hc, while GdMnO850 showed the highest value (~0.8), revealing that it is dominated by lower spatial dominant frequencies (Hc > 0.5) [21,23].

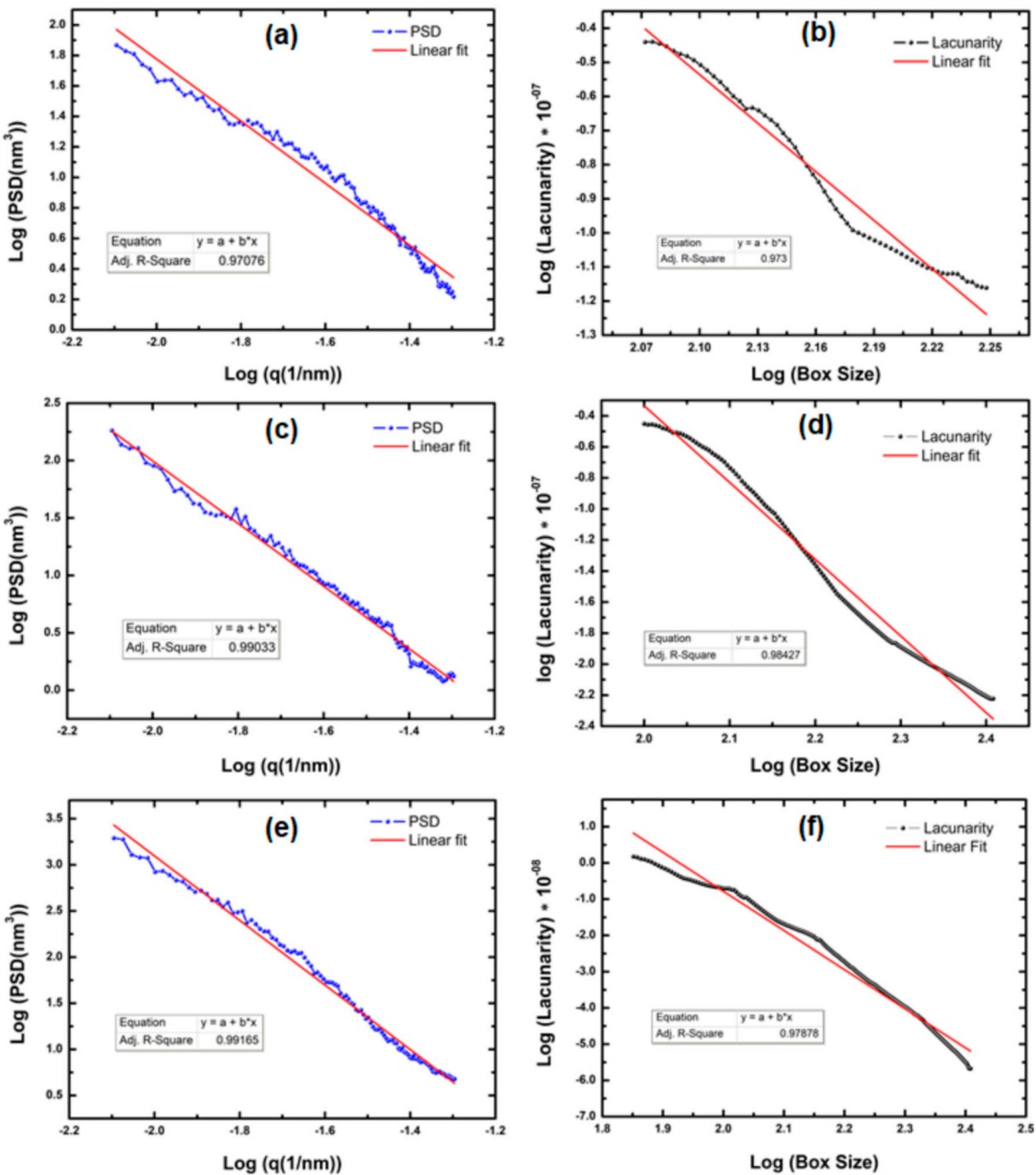

**Figure 5.** Surface lacunarity and power spectrum density (PSD) of (**a**,**d**) GdMnO650, (**b**,**e**) GdMnO750, and (**c**,**f**) GdMnO850, respectively.

**Table 5.** Fractal and parameters advanced fractals of GdMnO$_3$ thin films.

| Parameter | Unit | GdMnO650 | GdMnO750 | GdMnO850 |
|-----------|------|----------|----------|----------|
| FD | (-) | $2.242 \pm 0.034$ | $2.207 \pm 0.007$ | $2.281 \pm 0.025$ |
| HC | (-) | $0.169 \pm 0.172$ | $0.357 \pm 0.023$ | $0.813 \pm 0.038$ |
| $\lvert \beta \rvert$ | (-) | $4.02 \times 10^{-7} \pm 1.58 \times 10^{-7}$ | $4.09 \times 10^{-7} \pm 1.82 \times 10^{-7}$ | $8.45 \times 10^{-8} \pm 3.15 \times 10^{-8}$ |
| FS | (-) | $0.536 \pm 0.022$ | $0.500 \pm 0.024$ | $0.483 \pm 0.009$ |
| E | (-) | $0.962 \pm 0.019$ | $0.967 \pm 0.017$ | $0.996 \pm 0.003$ |

The power spectrum density showed that the films have different microtexture, which was already observed in stereometric parameters. Despite this, it was not possible to obtain accurate data on spatial complexity, homogeneity of surface microtexture, surface percolation, and topographic uniformity. However, the computed fractal parameters revealed that GdMnO850 has the highest spatial complexity because it has greater fractal dimension (~2.281), which is notably due to their roughness distribution. The lacunarity distribution curves of the sample's distribution fractal region obey the power law exposed in Equation (1), proving that the distribution of lacunes decreases persistently as a function of the box size. The values of the lacunarity exponents ($\lvert \beta \rvert$) shown in Table 5 show that GdMnO850 has the most homogeneous surface microtexture, as it exhibited the lowest $\lvert \beta \rvert$ value (~$8.5 \times 10^{-8}$), suggesting that this material is less likely to fail because of its fabrication process. The surface percolation is less for GdMnO850 (FS ~ 0.48) than for GdMnO650 (~0.54) and GdMn750 (~0.50), revealing that although GdMnO850 is assigned to be rougher, it is less probable that lower bands can be penetrated by fluid through film upper bands, in addition to having low surface percolation (FS < 0.5) [41] (Table 5). Finally, GdMnO850 also exhibited the highest surface entropy (0.996), demonstrating that this surface has greater topographic uniformity than any other sample, although all surfaces have exhibited E ~ 1 [48]. Generally, failures in materials based on ceramic systems are derived from non-uniformity and corrosion by erosion, where the penetration of fluids are associated with the erosion process and therefore better physical properties of these materials are requested. As our surface percolation for GdMnO850 was lower, we can suggest that in relation to the other samples, failures due to these types of problems are less likely.

In this paper, we have synthesized GdMnO$_3$ and deposited it onto Pt/TiO$_2$/SiO$_2$/Si substrates to obtain a correlation between fractal and stereometric parameters. A clear formation of the orthorhombic GdMnO$_3$ phase occurred for the film sintered at 850 °C, where a smaller grain size than for other systems previously reported was computed. In addition, a probable greater number of defects in relation to other systems studied previously was registered due to a higher value of lattice microstrain. Naturally, the structural arrangement affected the surface morphology because for films sintered at 650 and 750 °C the surface topography was smooth, while the film sintered at 850 °C had a rougher surface. Samples sintered at 850 °C presented the highest roughness, negative asymmetry, and flattened height distribution. Several other stereometric parameters such as functional, feature and hybrid demonstrated that the film sintered at 850 °C had the best topographical properties. Qualitative rendering revealed interesting aspects of the surface microtexture of the samples and confirmed that deeper valleys were assigned to be of the sample sintered at 850 °C. The smoother texture distribution of the film sintered at 850 °C indicates that this film may have more homogeneous microtexture. The stereometric results were confirmed by fractal and advanced fractal analysis because there was a correlation between these parameters. The results showed that the roughness distribution affected the Hurst coefficient and spatial complexity, where the film sintered at 850 °C reached the biggest values, as well as the emergence of topographic uniformity. Surface percolation and the lacunarity exponent of GdMnO$_3$ sintered at 850 °C were lower values, providing a surface with less surface percolation and more homogeneous microtexture, respectively.

Our results indicate that the process of deposition of $GdMnO_3$ thin films and the subsequent sintering at 850 °C produces films with interesting topographic properties that can be used to improve the process of fabrication of thin films based on systems involving rare earths, known as excellent multiferroic materials for technological application.

**Supplementary Materials:** The following are available online at https://www.mdpi.com/article/10.3390/app11093886/s1, Figure S1: SEM images of (a,b) GdMnO650, (c,d) GdMnO750, and (e,f) GdMnO850, Figure S2. 2D AFM topographic maps of (a,b) GdMnO650, (c,d) GdMnO750, and (e,f) GdMnO850., Figure S3. Graphical study of volume parameters (left) and Sk parameters (right) based upon the Abbott curve calculated for the samples: (a) GdMnO650, (b) GdMnO750, and (c) GdMnO850.

**Author Contributions:** Conceptualization, methodology, resources, J.P.d.C. and H.D.d.F.F.; data curation, original draft preparation, Y.R.-B.; writing—review and editing, supervision, R.S.M. and R.M.P.B.O.; funding acquisition, reviewing, Ş.Ţ.; formal analysis, investigation and project administration, J.A.M.; validation, reviewing, H.D.d.F.F. All authors have read and agreed to the published version of the manuscript.

**Funding:** This research received no external funding.

**Institutional Review Board Statement:** Not applicable.

**Informed Consent Statement:** Not applicable.

**Data Availability Statement:** The data presented in this study are available on request from the corresponding author. The data are not publicly available due to large amount of data.

**Acknowledgments:** The authors thank UFAM for the Analytical Center infrastructure.

**Conflicts of Interest:** The authors declare no conflict of interest. The funders had no role in the design of the study; in the collection, analyses, or interpretation of the data; in the writing of the manuscript, or in the decision to publish the results.

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
