# Peer review of "Fractal-Stereometric Correlation of Nanoscale Spatial Patterns of GdMnO3 Thin Films Deposited by Spin Coating"

_applsci, doi:10.3390/app11093886_

Round 1

Reviewer 1 Report

The use of multiferroic systems is a hot topic in thin-film fabrication. The authors have given a sound introduction to the research and presented the results thoroughly. However, the following items should be further addressed by the authors:

  1. Grammar issues exist in the article, for example, line 67 to 68. Please double-check them again.
  2. There are critical typos the authors should pay extra attention to. For example: line 19, GaMnO3;
  3. Figures in line 121, 128, 138, and 141 are in picture format. Please repopulate these equations using equation-building tools.
  4. The authors should address the issue that there is no control group. The sample pre-sintered at 400 C should be used as the control.

Reviewer 2 Report

As attached.

Reviewer 3 Report

In this research, authors investigated into the influence of processing parameters (such as temperature) on the surface morphology of spin-coated GaMnO3 thin films, by means of stereometric and fractal analysis. I suggest to accept the manuscript for publication, after authors addressed the following questions.

  • The scanning area used in this research is 2.5 by 2.5 µm2. How many AFM measurements have been done for each sample? How can authors ensure the scanned areas properly represent the whole surface morphology of each sample?
  • I suggest to include SEM images for each type of the sample in the revised manuscript.
  • Why the spin-coated thin films need to be pre-sintered in a tubular furnace for 9 times?
  • What is the thickness of the spin-coated thin films?

Round 2

Reviewer 1 Report

The authors have addressed all issues from my previous review. The manuscript might be accepted for publishing.

Author Response

Once again, thank the reviewer for the suggestions and considerations.

Reviewer 2 Report

As attached

Author Response

(The authors gave the same response as above.)

Reviewer 3 Report

I suggest to accept the manuscript for publication.

Author Response

(The authors gave the same response as above.)
